# Determinants of Left Atrial Compliance in the Metabolic Syndrome: Insights from the “Linosa Study”

**DOI:** 10.3390/jpm12071044

**Published:** 2022-06-27

**Authors:** Paolo Barbier, Edvige Palazzo Adriano, Daniela Lucini, Massimo Pagani, Gaspare Cusumano, Beatrice De Maria, Laura Adelaide Dalla Vecchia

**Affiliations:** 1Imaging Department, Jilin Heart Hospital, Changchun 130117, China; 2IRCCS Istituti Clinici Scientifici Maugeri, Department of Cardiology, 20138 Milan, Italy; edvige.palazzoadriano@icsmaugeri.it (E.P.A.); beatrice.demaria@icsmaugeri.it (B.D.M.); laura.dallavecchia@icsmaugeri.it (L.A.D.V.); 3BIOMETRA Department, University of Milan, 20122 Milan, Italy; daniela.lucini@unimi.it; 4Exercise Medicine Unit, IRCCS, Istituto Auxologico Italiano, 20135 Milan, Italy; massimo.paganiz@gmail.com; 5XI Reparto Mobile, Polizia di Stato, 20145 Palermo, Italy; cusumano.gaspare@gmail.com

**Keywords:** left atrial compliance, cardiovascular risk factors, metabolic syndrome, dyslipidaemia, echocardiography, cardiovascular prevention

## Abstract

The association between left atrial (LA) impairment and cardiovascular diseases (CVD) and between dyslipidaemia and CVD are well known. The present study aims to investigate the relationships between metabolic factors and LA dimensions and compliance, as well as test the hypothesis that metabolic factors influence LA function independent from hemodynamic mechanisms. Arterial blood pressure (BP), waist and hip circumference, metabolic indices, and a complete echocardiographic assessment were obtained from 148 selected inhabitants (M/F 89/59; age 20–86 years) of Linosa Island, who had no history of CVD. At enrollment, 27.7% of the subjects met the criteria for metabolic syndrome (MetS) and 15.5% for arterial hypertension (HTN). LA compliance was reduced in subjects with MetS compared to those without (53 ± 27% vs. 71 ± 29%, *p* = 0.04) and was even lower (32 ± 17%, *p* = 0.01) in those with MetS and HTN. At multiple regression analysis, the presence of MetS independently determined LA maximal area (r = 0.56, *p* < 0.001), whereas systolic BP and the total cholesterol/HDL cholesterol ratio determined LA compliance (r = 0.41, *p* < 0.001). In an apparently healthy population with a high prevalence of MetS, dyslipidaemia seems to independently influence LA compliance. At a 5-year follow-up, LA compliance was reduced in both all-cause and CVD mortality groups, and markedly impaired in those who died of CVD. These findings may contribute to understanding the prognostic role of LA function in CVD and strengthen the need for early and accurate lipid control strategies.

## 1. Introduction

The pathophysiology of left atrial (LA) involvement in cardiovascular disease (CVD) is complex. LA dilation often occurs secondary to left ventricular (LV) diastolic dysfunction and is characterized by progressive worsening along with the evolution of CVD [1]. LA remodeling has been suggested as an independent marker of LV hemodynamic burden in CVD [2,3]. Accordingly, population-based studies have shown that LA dimension can predict the presence of different CVDs [4,5,6] or related combined outcomes [7]. While LA systolic dysfunction occurs late and is secondary to LV structural remodeling and LA afterload mismatch [8], LA diastolic (reservoir) dysfunction occurs earlier [9,10]. LA diastolic filling, i.e., reservoir filling, is determined by both LA relaxation and compliance, the latter being a determinant of stroke volume [11,12] and an important determinant of LA function [2,12]. LA reservoir function as an index of LA compliance can be easily measured by echocardiography [2,11]. Concurrently, the evolving concept that the synergistic clustering of metabolic factors represents a significant CVD risk factor has fostered interest in the metabolic syndrome (MetS) [13,14,15,16,17,18] and its influence on LV function [13,19,20,21] and has given birth to the concept of cardiometabolic stress and risk [22]. Several studies have investigated a possible relation between LA properties and metabolic markers, such as obesity, insulin resistance, and the MetS [13,23,24,25], including the association between MetS and atrial fibrillation [26]. Such relationships would reinforce the central role of the LA chamber as a marker of CVD and related outcomes, although it is still unclear whether metabolic factors targeting vascular or cardiac tissues may cause LA dilation as a consequence of LV remodeling, or rather the thin LA wall itself represents a direct target. This study aimed to investigate the possible relationship between MetS and LA compliance and remodelling in the apparently healthy population of the “Linosa study” [27,28].

## 2. Materials and Methods

### 2.1. Study Population

The LINOSA study [27,28] enrolled the inhabitants of Linosa, a small (2.10 square miles) Mediterranean island in the Sicily Channel, for baseline screening. Out of 420 adult (≥18 y.o.) inhabitants, 364 individuals agreed to participate, and a complete data collection was available in 293. All participants were Caucasian. After exclusion of the patients with any history of chronic disease, 148 apparently healthy subjects were included in the present echocardiographic sub-study. None was on cardiovascular medications, and all gave informed consent. The study was conducted according to the guidelines of the Declaration of Helsinki and approved by the local Ethics Committee of the “Azienda Ospedaliera di Palermo” [27,28].

### 2.2. Clinical and Laboratory Data

Details about the methodology of data collection have been published elsewhere [27]. In the present study, the following data were collected: gender, age, history of cigarette smoking (current/previous/non-smoker), body surface area (BSA), body mass index (BMI), waist and hip circumference, waist/hip ratio, arterial blood pressure by brachial sphygmomanometer, and heart rate (HR) at rest. Laboratory parameters included blood glucose, total cholesterol (TC), low-density lipoprotein cholesterol (LDL-C), high-density lipoprotein cholesterol (HDL-C), TC/HDL-C ratio, triglycerides (TG), insulin and insulin sensitivity assessed by the homeostasis model (HOMA-IR) [29], and high sensitivity C-reactive protein (CRP). The MetS was diagnosed according to the NCEP-ATP III criteria [30] as the presence of three or more of the following features: hypertriglyceridemia >150.4 mg/dL; “low” HDL-C (<39.7 mg/dL in women and <40.15 mg/dL in men); hypertension (HTN) by systolic arterial pressure (SAP) >130 mmHg or diastolic arterial pressure > 85 mmHg; hyperglycaemia > 109.9 mg/dL; abdominal obesity by waist circumference > 88 cm in women and >102 cm in men. Obesity was defined as a BMI ≥ 35 Kg/m^2^. Further analysis was conducted on 4 subgroups identified by the separate or combined presence or absence of MetS and HTN [31]. Both all-cause and CVD mortality was recorded at a 5-year follow-up.

### 2.3. Echocardiographic and Carotid Ultrasound Examination

A standard 2D echocardiographic and color Doppler examination using a portable system SonoHeart Elite (SonoSite Inc, Basal, WA, USA) with a 4-2 MHz transducer was obtained for all subjects in the left lateral recumbent position. Images were acquired and stored digitally for later off-line analysis (average of 3 consecutive cycles) by 2 experienced cardiologists (LADV and PB). Measurements and calculations were obtained according to current guidelines [32]. More than trivial valvular regurgitation was excluded using color Doppler flow qualitative evaluation. From the parasternal long axis view, we measured M-mode LV end-diastolic diameter (LVEDD), LV end-systolic diameter (LVESD), end-diastolic and end-systolic posterior wall and interventricular septal thickness (PWd and PWs, IVSd and IVSs, respectively). We calculated: (i) end-diastolic hypertrophy index (EDHi), [(IVSd + PWd)/LVEDD]; (ii) LV mass index (LVMI), 0.8 × {1.04 × [LVEDD + PWd + IVSd)^3^ − LVEDD^3^]} + 0.6 indexed to body surface area (BSA) (gr/m^2^); (iii) fiber fractional shortening (FFS), [LVEDD − LVESD)/LVEDD] × 100 (%); (iv) LV end-systolic meridional wall stress (LVESS), 0.98 × (0.334 × SAP × LVESD)/[((PWs + IVSs)/2) × (1 + (((PWs + IVSs)/2)/LVESD)] − 2 (10^3^ dynes/cm^2^). LV diameters, volumes, and stroke volume were also indexed by BSA. Linear regression analysis was used to obtain FFS/LVESS curves as indexes of LV inotropic state. From the apical 4- and 2-chamber 2D views, biplane LV end-diastolic volume (LVEDV), LV end-systolic volume (LVESV), and LV ejection fraction (EF) were calculated using the modified Simpson’s method, and LV stroke volume (LVSV) was calculated as (LVEDV-LVESV) (ml). Effective arterial elastance (Ea) was calculated as a measure of aortic impedance (0.9 × SAP)/LVSV (mmHg/mL). Monoplane 4-chamber maximal and minimal LA areas indexed to BSA (LAmaxi and LAmini, respectively) were measured. LA reservoir function (LARes), [(LAmaxi − LAmini)/LAmini] was used as an index of LA compliance [11]; a cut-off value of 50% was chosen to differentiate normal (≥50%) from abnormal (< 50%) LA compliance [33]. From the Doppler examination, we measured the peak early (Ep) and late (Ap) diastolic mitral flow velocity (cm/s) and velocity–time integrals (Ei and Ai, cm) and the deceleration time (Edt, ms) of Ep. We calculated: Ep/Ap ratio, and the early {[Ei/(Ei + Ai)] × 100, (%)} and late {[Ai/(Ei + Ai)] × 100 (%)} LV fractional filling ratios as indices of LV filling during LV relaxation and LA systolic function, respectively. The intima-media thickness of the common carotid artery (IMT) was measured by B-mode vascular ultrasound (SonoHeart Elite, SonoSite Inc, WA, USA with an 8 MHz transducer). Interobserver variability of LAmaxi and LVEDV measurements was obtained in 20 randomly selected patients, using Bland–Altman analysis. No correlation was found between means and differences for the variables analysed. The SD of the differences was 1.8 cm^2^ for LAmax and 11 mL for the LVEDV.

### 2.4. Statistical Analysis

Continuous data are reported as mean ± SD. The unpaired t-test (or Mann–Whitney rank sum test when appropriate) was used to test for differences between patients with and without MetS. One-way ANOVA (Holm–Sidak test for multiple comparisons) or KruskalWallis one-way ANOVA on ranks (Dunnett test for multiple comparisons), in case of not-normal distribution, were used to test for differences between normal subjects and patients with either MetS or HTN, and those with both MetS and HTN. One-way ANOVA (Holm–Sidak test for multiple comparisons versus the group of alive patients), or Kruskal–Wallis one-way ANOVA on ranks (Dunnett test for multiple comparisons versus the group of alive patients) in case of not-normal distribution, were used to test for differences between mortality groups. Stepwise multiple regression analysis was used to detect independent determinants of LA compliance and dimensions, LA systolic function, and LV mass and diastolic function. A *p* < 0.05 was considered significant. Statistical analysis was performed using SPSS software (SPSS Inc., Chicago, IL, USA), version 20.

## 3. Results

### 3.1. Metabolic and Echocardiographic Parameters

Table 1 summarizes the main clinical characteristics and laboratory tests for all participants, and for those without and with MetS. Criteria for MetS were met in 41 subjects (27.7%).

Altogether, males had higher SAP (134 ± 18 vs. 123 ± 19 mmHg, *p* = 0.001), lower HR (66 ± 10 vs. 73 ± 11 bpm, *p* < 0.001), lower HDL-C (38 ± 10 vs. 45 ± 10 mg/dL, *p* < 0.001), higher TG (123 ± 62 vs. 99 ± 50 mg/dL, *p* < 0.001), and higher waist/hip ratio (0.92 ± 0.06 vs. 0.87 ± 0.09, *p* < 0.001) than females. 

The prevalence of MetS was 27.7 % (regardless of sex, HR, and CRP), increasing with age (i.e., 17% in subjects <40 years, 40% in the 40–59 age range, and 43% in subjects >59 years), while all other characteristic variables were consistently matched (Table 1).

Table 2 summarizes the main echocardiographic and vascular ultrasound findings.

Figure 1 shows that in subjects with MetS, LA compliance was significantly lower in MetS patients than in subjects without MetS. 

LV dimensions, systolic and diastolic function indices, IMT and Ea were within normal limits. Of note, LV filling occurred predominantly during early diastole, a finding consistent with normal LV relaxation alongside normal LV dimensions and systolic function, even though LVMI was increased, mainly due to eccentric LV hypertrophy (H) (75% of the subjects, of whom 95% were obese) (Figure 2).

At multiple regression analysis, the determinants of LVMI (r = 0.62, *p* < 0.001) were the waist/hip ratio (B = 161, SE = 48, *p* = 0.001), DAP (B = 1.7, SE = 0.38, *p* < 0.001), and HR (B = −1, SE = 0.34, *p* = 0.003), whereas the independent predictors of E/A (r = 0.76, *p* < 0.001) were age (B = −0.014, SE = 0.002, *p* < 0.001), HR (B = −0.016, SE = 0.004, *p* < 0.001), and the presence of MetS (B = −0.24, SE = 0.09, *p* = 0.01). When the MetS factor was excluded from analysis, hypertriglyceridemia (>50 mg/dL) entered the equation (B = −0.24, SE = 0.09, *p* = 0.01). Both Ep and Edt were determined exclusively and mutually by age (r = 0.37, *p* < 0.001 and r = 0.43, *p* < 0.001, respectively).

Compared to subjects without MetS, Ea, LVESS, and LVMI were higher in the subgroup of subjects with MetS (Table 2), whereas LV contractility indices were similar (Figure 3).

Different from those without MetS, in subjects with MetS, early filling was decreased and followed by a compensatory increase in late filling, consistent with diastolic dysfunction due to prolonged relaxation (Table 2).

### 3.2. Left Atrial Anatomy and Function

LA reservoir function was reduced (<50%) in 26% of the subjects. These subjects were older (52 ± 19 vs. 42 ± 15 years, *p* < 0.01), with higher SAP (137 ± 22 vs. 127 ± 17 mmHg, *p* = 0.009), higher prevalence of MetS (47% vs. 21, *p* < 0.01), higher TC (219 ± 42 vs. 197 ± 37 mg/dL, *p* = 0.008), higher LDL-C (157 ± 39 vs. 138 ± 34 mg/dL, *p* = 0.01), greater LAmaxi (10 ± 2 vs. 8.8 ± 2 cm/m^2^, *p* = 0.002), longer Edt (184 ± 53 vs. 164 ± 35 ms, *p* = 0.02), and higher IMT (0.61 ± 0.01 vs. 0.56 ± 0.01, *p* = 0.02), compared to subjects with normal LA reservoir function.

At univariate analysis performed in the whole population, we found mild positive correlations between both LAmaxi and LAmini and SAP (r = 0.35, *p* < 0.001 and r = 0.23, *p* = 0.008) and mild negative correlations between LA reservoir function and both SAP (r = −0.32, *p* < 0.001) and LVESS (r = −0.23, *p* = 0.009). Different from LV diastolic function indices, LA variables were not related to HR or age. A minor correlation was also found between the lipid profile and both LAmaxi and LAmini (TC, r = 0.29, *p* = 0.001; r = 0.22, *p* = 0.01. LDL-C, r = 0.28, *p* = 0.001; r = 0.21, *p* = 0.02), as well as LA reservoir function (TC, r = −0.24, *p* = 0.007; LDL-C, r = −0.24, *p* = 0.008; TC/HDL-C, r = −0.26, *p* < 0.001; triglycerides, r = −0.23, *p* = 0.01). Finally, LA reservoir function was not related to the LV geometry pattern (Figure 2).

Although still within normal limits, LAmaxi was larger in subjects with MetS than in subjects without MetS (Figure 1). In contrast, LA reservoir function was significantly reduced and abnormal in 58% of subjects with MetS (χ^2^, *p* = 0.007), whereas it was normal (>50%) in 82% of subjects without MetS. At multiple regression analysis, LAmaxi was determined (r = 0.56, *p* < 0.001) by the presence of MetS (B, 0.99, SE, 0.04, *p* = 0.007), LVEDV (B, 0.08, SE, 0.01; *p* < 0.001), and sex (B, −0.7, SE, 0.33, *p* = 0.04). After exclusion of MetS, TC (B = 0.008, SE = 0.004, *p* = 0.03) entered the equation (r = 0.53, *p* < 0.001). The LA reservoir function was predicted (r = 0.41, *p* < 0.001) by SAP (B = −0.54, SE = 0.2, *p* = 0.001) and by TC/HDL-C ratio (B = −4.2, SE = 1.8, *p* = 0.02). In contrast, LA systolic function was determined (r = 0.47, *p* < 0.001) by age (BE = 0.24, SE = 0.05, *p* < 0.001) and HR (B = 0.2, SE = 0.07, *p* = 0.007), as expected.

### 3.3. Subgroup Analysis

Based on the presence of MetS (41 subjects, 27.7%) and HTN (23 subjects, 15.5%), four subgroups were identified: Group 1, subjects without MetS and normotensive; Group 2, subjects with MetS and normotensive; Group 3, subjects without MetS but hypertensive and Group 4, subjects with both MetS and hypertension (Table 3). 

Prevalence of obesity was 7.9% in Group 1, 10.3% in Group 2, 23.8% in Group 3, and 66.7% in Group 4. Compared to Group 1, subjects in Group 2 were one decade older and mostly male, whereas subjects in Group 4 were three decades older and mostly male. 

Table 4 summarizes the main echocardiographic and vascular ultrasound findings in the four subgroups.

Of note, all groups were characterized by some degree of LV hypertrophy (LVH). In Groups 1 and 2, despite normal blood pressure, LVH (mostly eccentric) was found in 79% and 76% of subjects, respectively. In Group 3, eccentric LVH was found in 67% of subjects, whereas in Group 4, eccentric LVH was present in 92%, together with increased LVESS but normal aortic impedance and LV systolic function. In both Groups 1 and 3, Ea, IMT, and both LV and LA structure and function were normal. In contrast, in Group 2, LAres (LA compliance) was impaired despite normal LV diastolic function and LA dimensions (Figure 4), whereas IMT was increased compared to Group 1. In Group 4, LAres was decreased even further, despite normal LAmaxi, together with increased IMT and higher Ep/Ap. The latter resulted from a decrease in LV early filling and a compensatory increase in the late filling.

### 3.4. Mortality at 5 Years

Mortality data were acquired at a 5-year follow-up through the general practitioner in Linosa: 20 subjects (13.5%, M/F 10/10, age 53 ± 15 years) had died, of whom 4 (2.7%, M/F 3/1, age 51 ± 13 years) died from CVD. See Table 5 for the details. 

Compared to surviving subjects, LA reservoir function was decreased in both all-cause and CVD mortality groups, of note markedly impaired in those who died from CVD. The only different metabolic factor in the group comparison was HDL-C, which was reduced in both all-cause and CVD deaths (Table 5). Subjects who died of CVD also showed a significantly higher LVMI and LVEDVI, associated with lower LVEF. Interestingly, all these subjects were normotensive, three of them had MetS (one male was in Group 1, two males and one female were in Group 2), suggesting a strong negative prognostic significance of LA dysfunction. 

## 4. Discussion

In a homogeneous population without known CVD, decreased LA compliance (an index of LA diastolic dysfunction) was frequent and independently related to the presence of dyslipidaemia and MetS, which had a high prevalence (27.7%), in agreement with previous studies [25]. 

In the isolated and small Caucasian population of the Linosa study, the prevalence of MetS was higher than in the general Italian population (27% vs. 15–22%) [34]. The MetS, a cluster of lipid and non-lipid factors, is considered an independent CVD risk factor [15,17,18]. It is still unclear whether it represents the result of the synergistic interaction, or merely the sum, of its constituent factors [14,22]. In any case, the concept of cardiometabolic risk [22] implies that metabolic factors may cause, directly or indirectly, cardiac dysfunction. Impaired LA compliance represents an early sign of cardiac disease and may contribute together with the metabolic factors to increase CVD risk. Of note, the LA chamber can be easily, non-invasively, and reliably assessed by echocardiography.

### 4.1. The Metabolic Syndrome and LV Diastolic Dysfunction

In this study, subjects with MetS showed an increased LVMI, despite normal blood pressure and arterial elastance. These subjects were characterized by a prolonged relaxation type of diastolic dysfunction with a shift from early to late diastolic filling. However, MetS (and hypertriglyceridemia) was an independent determinant of the LV filling pattern, regardless of the LV mass, as also suggested in previous studies [21,35,36,37,38,39,40,41,42]. Several mechanisms may cause LV dysfunction in MetS, such as pressure and volume overload, LV myocyte dysfunction, insulin resistance, central obesity, and altered lipid profile [20,23,35,43,44]. Even mild or borderline HTN, frequent in the MetS [16,17,25,36], may prolong LV relaxation, thus causing LV diastolic dysfunction. Abdominal adipose tissue enhances endothelial dysfunction, arterial stiffness, LV diastolic dysfunction, and LV remodeling [44]. Insulin resistance also increases arterial stiffness, and elevated circulating free fatty acids have toxic effects on the LV myocardium [37]. A cholesterol-rich diet may induce a “cholesterol cardiomyopathy” in rabbits characterized by LV systolic and relaxation dysfunction (through altered intracellular calcium handling) [20]. A direct, LV mass independent, negative effect of TC and LDL-C on LV diastolic function has been suggested in untreated mild-to-moderate hypertensive postmenopausal women [39]. In treated hypertensive patients, low HDL-C has been identified as an independent predictor of both LVH (independent of blood pressure and age) [39,40] and LV diastolic dysfunction [39]. Overall, although elevated blood pressure is needed for the development of pre-clinical CVD, metabolic risk factors amplify its effects [14]. 

### 4.2. The Metabolic Syndrome and Left Atrial Remodeling and Function

LA remodeling has gained acceptance as an easy-to-measure marker of CVD [6], predicting combined outcomes in the general population [2,3,7]. Because of its crucial position upstream to the left ventricle, the LA chamber represents both a target of LV anatomical and functional remodeling and a modulator between LV diastolic pressures and pulmonary wedge pressure, explaining its diagnostic and prognostic roles. Either a reduced LV early filling caused by LV hypertrophy or an increased LV diastolic pressure caused by increased LV stiffness may induce progressive LA dilation in chronic heart diseases [41]. From a diagnostic point of view, compensatory LA remodeling may be used as a simple “LV pressure and volume barometer” to detect LV remodeling and dysfunction. 

The main cardiac functional alteration found in the subjects with MetS of this study was an impaired LA compliance (LA diastolic dysfunction), which was worse in subjects with combined MetS and HTN, interestingly not associated with major LA remodeling. Furthermore, the main determinants of the reduced LA compliance were increased arterial pressure and an altered lipid profile, independent from LV hypertrophy and LV diastolic dysfunction. 

Of note, the suggested effect of the altered lipid profile on LV diastolic function was quantitatively less than that observed on LA compliance. Importantly, the hypothesis of a direct metabolic effect on the left atrium is strengthened by the fact that LA remodeling (secondary to LV diastolic dysfunction) was not significant in the subjects of this study. Indeed, LA compliance is generally not directly influenced by LV diastolic dysfunction [11].

Given the “barometer” role of the left atrium to LV haemodynamics, some degree of LA remodeling is expected in patients with LV hypertrophy or systolic and diastolic dysfunction secondary to MetS or its constituent factors (namely, obesity and mild arterial hypertension) [23,34]. Early experimental data have shown extensive LA endocardial lesions in rats fed with a high-fat, low-protein diet, suggesting a direct effect of lipid overload on the LA wall [45] and supporting the hypothesis of a direct relation between the lipid profile and LA dysfunction. Thereafter, few studies have found an independent influence of selected metabolic factors on LA dilation: obesity in hypertensive men [46], increased glucose intolerance across HOMA quartiles in both sexes, independent from LV mass [47], and the MetS in uncomplicated hypertensives [25] and adolescents [13]. Accordingly, our study suggests a direct effect of the MetS and specifically of the lipid profile on LA compliance and extends these findings by providing new evidence of significant LA diastolic dysfunction secondary to metabolic factors related to the MetS, independently from LV remodeling and diastolic dysfunction. 

To strengthen our conclusions, the subgroup analysis suggests that HTN by itself had no significant effect on LA structure and compliance, whereas it had a negative synergistic effect when associated with the MetS. These results further reinforce the current strategies of strict lipid control management to reduce cardiovascular risk, especially in patients with higher risk factors [48]. 

Finally, although no conclusions could be drawn concerning CVD mortality because of the small sample size and the low number of events, the mortality data at 5 years may suggest the negative prognostic value of LA diastolic dysfunction, regardless of the presence of MetS or HTN.

### 4.3. Limitations

Although the sample population analysed was small, it did represent a unique closed population free from confounders. Linosa is a small Mediterranean island, which has been “genetically” isolated since the Borboni sent over the original settlers in 1820. Even today, despite being a tourist destination during the summer months, doubling or tripling the residents, for the rest of the year it remains an isolated island, with few connections possible only by sea or, in case of emergency, by helicopter. It should be kept in mind that the high prevalence of MetS in this sample population might amplify the effects of metabolic factors on LA structure and function. In this respect, further studies in larger populations are needed to confirm these results. The insights of mortality analysis should also be confirmed in a larger population. In addition, LA pressure, which influences both LA structure and function, could not be estimated, since tissue Doppler technology was not available on the portable ultrasound equipment used in this study. However, given the fact that all participants were asymptomatic, with no history of effort dyspnoea, and that LA dimensions, LV structure, and systolic function were normal, we can reasonably assume normal LV filling pressures and normal diastolic function. Given such premises, although LV diastolic function could not be estimated according to current guidelines, our assessment, based on the trasmitral flow profile (velocity–time integrals), although limited, can be reasonable in the population of this study without structural LV and LA abnormalities. Current guidelines also suggest the estimation of LA compliance with the use of longitudinal strain technology, which was not available at the time of the Linosa study. However, LA compliance was evaluated based on a validated method (11). Unfortunately, only mortality data, but not the cardiovascular morbidity, could be retrieved at follow-up. The low number of cardiovascular deaths could only suggest the association between atrial dysfunction and increased cardiovascular risk.

## 5. Conclusions

This study on an apparently healthy population characterized by low genetic and behavioral variance but elevated prevalence of metabolic syndrome suggests a direct independent influence of metabolic factors—in particular the lipid profile—on LA compliance. These relationships may help explain the recently established important prognostic role of the left atrium in CVD and strengthen the need for accurate and strict lipid control strategies. Given the small sample and the closed population, further studies in larger populations are needed to confirm these preliminary results.

## Figures and Tables

**Figure 1 jpm-12-01044-f001:**
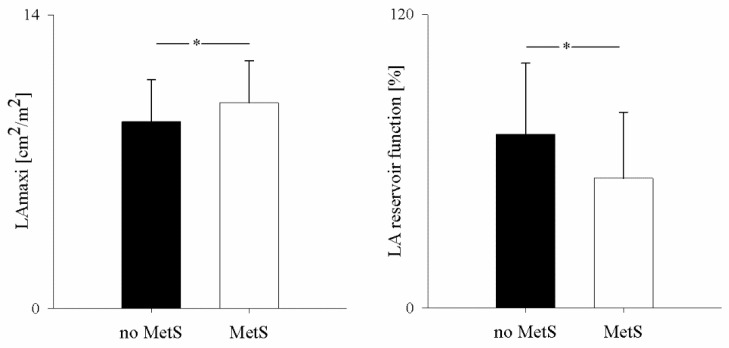
Left panel: Comparison of indexed monoplane 4-chamber maximal left atrial area (LAmaxi) in subjects without Metabolic Syndrome (no MetS) (black bars) and with MetS (MetS) (white bars); right panel: comparison of LA reservoir function in no MetS (black bars) and MetS subjects (white bars); * *p* < 0.01 Mets vs. no MetS.

**Figure 2 jpm-12-01044-f002:**
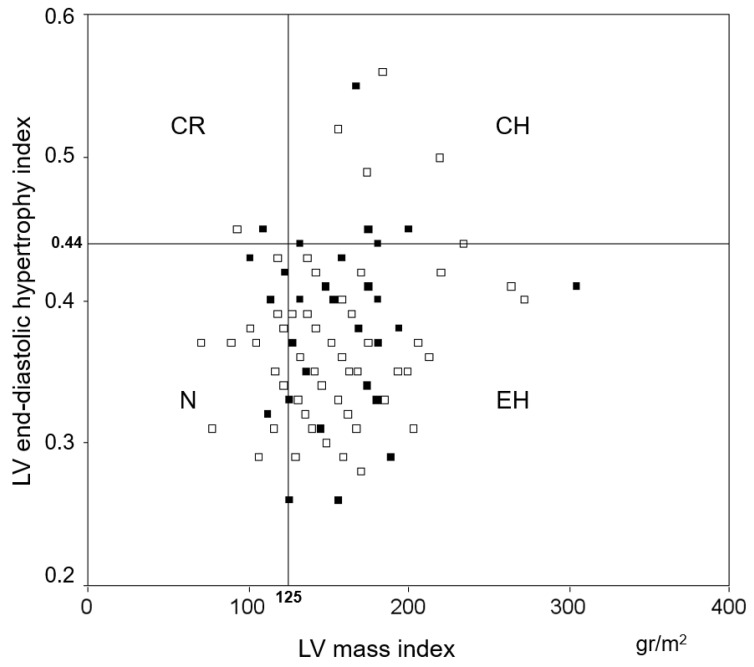
Distribution of left ventricular (LV) geometry in the whole population. The majority of patients is positioned in the LV eccentric hypertrophy quadrant (75%). The absence of a relation between LV geometry and LA compliance can also be appreciated (empty squares: normal LA compliance; filled squares: reduced LA compliance). CH: concentric hypertrophy; CR: concentric remodeling; EH: eccentric hypertrophy; LA: left atrial; N: normal.

**Figure 3 jpm-12-01044-f003:**
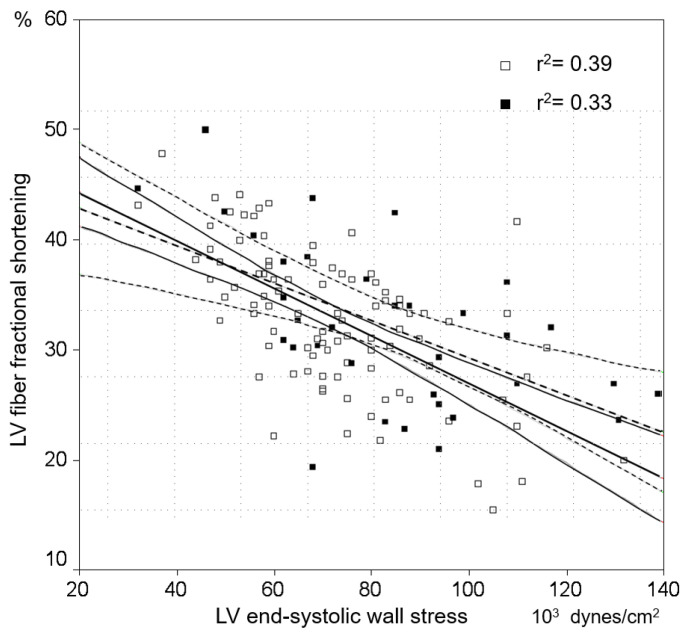
Distribution of left ventricular (LV) geometry in the whole population. The majority of patients is positioned in the LV eccentric hypertrophy quadrant (75%). The absence of a relation between LV geometry and LA compliance can also be appreciated (empty squares: normal LA compliance; filled squares: reduced LA compliance). CH: concentric hypertrophy; CR: concentric remodeling; EH: eccentric hypertrophy; LA: left atrial; N: normal.

**Figure 4 jpm-12-01044-f004:**
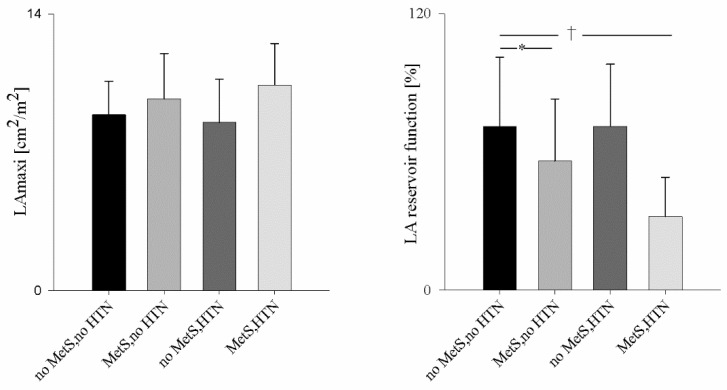
Left panel: Comparison of indexed monoplane 4-chamber maximal left atrial area (LAmaxi) in subjects without Metabolic Syndrome (no MetS) and normotensive (no HTN) (black bars), subjects with MetS and no HTN (grey bars), subjects no MetS but hypertensive (HTN) (dark grey bars), and subjects with MetS and HTN (light grey bars); right panel: comparison of LA reservoir function in no MetS and no HTN (black bars), MetS and no HTN (grey bars), no MetS but HTN (dark grey bars), and MetS and HTN (light grey bars); * *p* < 0.001, Mets, no HTN vs. no MetS, no HTN; † *p* < 0.001, MetS, HTN vs. no MetS, no HTN.

**Table 1 jpm-12-01044-t001:** Clinical characteristics for all enrolled subjects (All), those without metabolic syndrome (no MetS), and those with MetS (MetS).

	All *n* = 148	no MetS *n* = 107 (72%)	MetS *n* = 41 (28%)
Age, years	44.6 ± 16	41.3 ± 15	54.6 ± 16 **
Gender, M/F	89/59	62/45	27/14
BSA, m^2^	1.83 ± 0.22	1.81 ± 0.23	1.93 ± 0.15 **
BMI, Kg/m^2^	26.3 ± 4.4	25.6 ± 4.5	28.2 ± 3.6 *
Waist/Hip ratio	0.91 ± 0.01	0.89 ± 0.01	0.95 ± 0.04 *
HR, bpm	69 ± 11	69 ± 11	67 ± 9
SAP, mmHg	130 ± 19	125 ± 17	145 ± 19 **
DAP, mmHg	75 ± 9	73 ± 9	79 ± 9 **
IMT	0.57 ± 0.09	0.54 ± 0.07	0.64 ± 0.1 **
HOMA-IR, a.u.	1.8 ± 1.39	1.4 ± 0.65	2.6 ± 2.2 **
TC, mg/dL	203 ± 40	194 ± 35	228 ± 41 **
HDL-C, mg/dL	41 ± 11	43 ± 11	35 ± 8 **
LDL-C, mg/dL	143 ± 36	135 ± 33	163 ± 35 **
C/HDL-C ratio	5.2 ± 1.6	4.7 ± 1.2	6.8 ± 1.5 **
TG, mg/dL	114 ± 58	89 ± 32	185 ± 58 **
CRP, mg/dL	2.1 ± 2.5	2.2 ± 2.6	1.8 ± 2.1
Glycemia, mg/dL	87 ± 13	83 ± 9	96 ± 17 **

a.u.: arbitrary units; BSA: body surface area; BMI: body mass index; CRP: C-reactive protein; DAP: diastolic arterial pressure; HDL-C: high-density lipoprotein cholesterol; HOMA-IR: homeostatic model; HR: heart rate; IMT: carotid intima-media thickness; LDL-C: low-density lipoprotein cholesterol; SAP: systolic arterial pressure; TC: total cholesterol; TG: triglycerides. * *p* < 0.01, w/Mets vs. w/o MetS; ** *p* < 0.001, w/Mets vs. w/o MetS.

**Table 2 jpm-12-01044-t002:** Echocardiographic and vascular ultrasound variables in all subjects (All), subjects without metabolic syndrome (no MetS), and with MetS (MetS).

	All *n* = 148	no MetS *n* = 107 (72%)	MetS *n* = 41 (28%)
Ea, mmHg/mL	1.85 ± 0.44	1.81 ± 0.42	2.04 ± 0.44 *
LVEDDI, cm/m^2^	2.7 ± 0.3	2.7 ± 0.3	2.6 ± 0.3
LVFFS, %	33 ± 6.9	33 ± 6.7	32 ± 7.4
LVPWd, cm	0.9 ± 0.1	0.91 ± 0.1	0.94 ± 0.1
IVSd, cm	0.9 ± 0.1	0.87 ± 0.1	0.97 ± 0.1
LVEDHI, cm/m^2^	0.37 ± 0.05	0.37 ± 0.05	0.39 ± 0.04
LVMI, gr/m^2^	153 ± 36	148 ± 32	171 ± 44 **
LVESS, 10^3^ dynes/cm^2^	76 ± 22	72 ± 19	84 ± 25 *
LVEDVI, ml/m^2^	59 ± 13	59 ± 12	61 ± 14
LVEF	61 ± 6	62 ± 6	58 ± 7 *
LVSVI, mL/m^2^	36 ± 8	36 ± 8	35 ± 8
Ep, cm/s	65 ± 17	68 ± 14	60 ± 21 *
Edt, ms	169 ± 40	166 ± 40	177 ± 44
Ap, cm/s	52 ± 16	51 ± 15	57 ± 20
Ep/Ap	1.3 ± 0.4	1.4 ± 0.4	1.2 ± 0.5 *
E fractional filling, %	63 ± 9	64 ± 9	60 ± 9 *
A fractional filling, %	37 ± 9	36 ± 9	40 ± 9 *
IMT, mm	0.57 ± 0.09	0.54 ± 0.07	0.64 ± 0.1

Ap: mitral peak A wave; Ea: Effective arterial elastance; Edt: mitral E wave deceleration time; Ep: mitral peak E wave; IMT: carotid intima-media thickness; IVSd: end-diastolic interventricular septum thickness; LAmaxi: indexed monoplane 4-chamber maximal left atrial area; LV: left ventricular; LVEDDI_:_ indexed end-diastolic diameter; LVEDHI: end-diastolic hypertrophy index; LVEDVI: end-diastolic volume index; LVEF: ejection fraction; LVESS: end-systolic stress; LVFFS: fiber fractional shortening; LVMI: mass index; LVPWd: end-diastolic posterolateral wall thickness; LVSVI: LV stroke volume index; MetS: metabolic syndrome. * *p* < 0.01 Mets vs. No MetS; ** *p* < 0.001 MetS vs. No MetS.

**Table 3 jpm-12-01044-t003:** Patient demographics, hemodynamic, and metabolic variables in the four subgroups defined by the absence/presence of metabolic syndrome (no MetS and MetS, respectively) and absence/presence of hypertension (no HTN and HTN, respectively).

	Group 1 no MetS, no HTN *n* = 90	Group 2 MetS, no HTN *n* = 35	Group 3 no MetS, HTN *n* = 7	Group 4 MetS, HTN *n* = 16
Age, years	41 ± 14	51 ± 15 *	42 ± 18	77 ± 12 ††
Gender, M/F	44/46	29/5	5/2	11/6
BMI, Kg/m^2^	25.7 ± 4.9	27.8 ± 3.3	25.8 ± 2.8	30 ± 4.9 †
Waist/Hip ratio	0.89 ± 0.08	0.95 ± 0.04 **	0.91 ± 0.07	0.96 ± 0.04 †
HR, bpm	70 ± 11	68 ± 9	68 ± 12	64 ± 7
SAP, mmHg	124 ± 17	145 ± 20 **	129 ± 15	150 ± 9 †
DAP, mmHg	73 ± 10	79 ± 8 *	74 ± 9	75 ± 7
HOMA-IR, a.u.	1.5 ± 0.7	2.8 ± 2.4 **	1.4 ± 0.6	3 ± 1.2 †
TC, mg/dL	197 ± 35	222 ± 36 *	184 ± 37	255 ± 59 ††
HDL-C, mg/dL	44 ± 11	34 ± 8 **	41 ± 8	40 ± 8
LDL-C, mg/dL	137 ± 33	159 ± 32 *	129 ± 34	182 ± 45 †
TC/HDL ratio	4.7 ± 1.2	6.9 ± 1.6 **	4.6 ± 1.2	6.4 ± 0.8 †
TG, mg/dL	91 ± 34	188 ± 62 **	81 ± 25	174 ± 40 ††
CRP, mg/dL	2.1 ± 2.4	1.9 ± 2.3	3 ± 3.2	1.4 ± 0.4
Glycemia, mg/dL	85 ± 8	94 ± 17 **	79 ± 9	107 ± 12 ††

a.u.: arbitrary units; BMI: body mass index; CRP: C-reactive protein; DAP: diastolic arterial pressure; HDL-C: high-density lipoprotein cholesterol; HOMA-IR: homeostatic model; HR: heart rate; HTN: arterial hypertension; LDL-C: low-density lipoprotein cholesterol; MetS: metabolic syndrome; SAP: systolic arterial pressure; TC: total cholesterol; TG: triglycerides. * *p* < 0.01 and ** *p* < 0.001, Group 2 vs. 1; † *p* < 0.05 and †† *p* < 0.001, Group 4 vs. 1.

**Table 4 jpm-12-01044-t004:** Patient echocardiographic and vascular ultrasound variables in the four subgroups defined by the absence/presence of metabolic syndrome (no MetS and MetS, respectively) and absence/presence of hypertension (no HTN and HTN, respectively).

	Group 1 No MetS, No HTN *n* = 90	Group 2 MetS, No HTN *n* = 35	Group 3 No MetS, HTN *n* = 7	Group 4 MetS, HTN *n* = 16
Ea, mmHg/mL	1.79 ± 0.44	2.06 ± 0.44 *	1.88 ± 0.37	1.91 ± 0.46
LVEDDI, cm/m^2^	2.7 ± 0.3	2.6 ± 0.3	2.7 ± 0.4	2.7 ± 0.3
LVPWd, cm	0.9 ± 0.1	0.95 ± 0.1	0.92 ± 0.1	0.9 ± 0.1
IVSd, cm	0.87 ± 0.1	0.98 ± 0.1 **	0.88 ± 0.1	0.93 ± 0.1
LVEDHI	0.37 ± 0.06	0.39 ± 0.04	0.37 ± 0.05	0.38 ± 0.04
LVMI, g/m^2^	147 ± 33	175 ± 47 *	151 ± 28	155 ± 22
LVESS, 10^3^ dynes/cm^2^	72 ± 11	82 ± 23	75 ± 19	97 ± 33 †
LVEDVI, mL/m^2^	59 ± 12	59 ± 13	57 ± 13	67 ± 16
LVEF, %	62 ± 6	57 ± 7 *	63 ± 5	61 ± 5
LVSVI, mL/m2	36 ± 8	34 ± 7	36 ± 9	41 ± 10
Ep/Ap	1.4 ± 0.4	1.2 ± 0.5	1.4 ± 0.5	0.8 ± 0.2 †
E fractional filling, %	64 ± 8	61 ± 8	66 ± 10	53 ± 12 †
A fractional filling, %	36 ± 8	39 ± 8	34 ± 10	47 ± 12 †
IMT, mm	0.54 ± 0.07	0.64 ± 0.11 **	0.55 ± 0.07	0.68 ± 0.05 ††

Ap: mitral peak A wave; Ea: Effective arterial elastance; Edt: mitral E wave deceleration time; Ep: mitral peak E wave; IMT: carotid intima media thickness; IVSd: end-diastolic interventricular septum thickness; LV: left ventricular; LVEDDI_:_ LV indexed end-diastolic diameter; LVEDHI: LV end-diastolic hypertrophy index; LVEDVI: end-diastolic volume index; LVEF: LV ejection fraction; LVFFS: LV fiber fractional shortening; LVESS: LV end-systolic stress; LVMI: LV mass index; LVPWd: LV end-diastolic posterolateral wall thickness; LVSVI: LV stroke volume index; MetS: metabolic syndrome. * *p* < 0.01 and ** *p* < 0.001, Group 2 vs. 1; † *p* < 0.05 and †† *p* < 0.001, Group 4 vs. 1.

**Table 5 jpm-12-01044-t005:** Main clinical and echocardiographic characteristics at a 5-year follow-up.

	Alive *n* = 128	CV Deaths *n* = 4	All-Cause Deaths *n* = 20
Age, years	44 ± 16	51 ± 13	53 ± 15 **
Gender, M/F	79/49	3/1	10/10
BMI, Kg/m^2^	26.3 ± 4.5	27.4 ± 2.2	27.8 ± 4.4
HR, bpm	69.1 ± 10.7	64.3 ± 6.8	69.4 ± 10.2
SAP, mmHg	129.6 ± 19.3	135.3 ± 19.8	136.5 ± 20.5
DAP, mmHg	74.6 ± 9.3	74.5 ± 9.0	76.6 ± 8.5
HOMA-IR, a.u.	1.8 ± 1.4	2.2 ± 0.5	2.1 ± 1.2
TC, mg/dL	202.2 ± 39.8	221.7 ± 48.1	206.6 ± 42.3
HDL-C, mg/dL	41.1 ± 10.7	32.0 ± 3.5 *	35.2 ± 8.8 *
LDL-C, mg/dL	142.3 ± 35.8	159.7 ± 42.1	152.0 ± 39.7
TG, mg/dL	112.4 ± 57.3	175.7 ± 73.7	135.4 ± 55.7
Glycemia, mg/dL	87.0 ± 12.8	85.5 ± 7.1	91.0 ± 10.2 *
LVPWd, cm	0.9 ± 0.1	1.0 ± 0.1	1.0 ± 0.1
IVSd, cm	0.9 ± 0.1	1.0 ± 0.1 *	1.0 ± 0.1 *
LVMI, gr/m^2^	151.9 ± 34.3	206.2 ± 66.9 *	169.2 ± 46.3
LVESS, 10^3^ dynes/cm^2^	75.9 ± 21.6	77.7 ± 27.8	73.3 ± 18.2
LVEDVi, ml/m^2^	58.8 ± 12.0	79.3 ± 16.6 *	63.8 ± 13.2
LVEF, %	61.0 ± 6.3	51.0 ± 7.1 *	59.3 ± 7.2
LVSVI, ml/m^2^	35.8 ± 8.3	39.7 ± 3.4	37.2 ± 4.9
Ep/Ap	1.3 ± 0.4	1.8 ± 0.7	1.3 ± 0.5
E fractional filling, %	62.9 ± 9.1	69.4 ± 11.6	62.2 ± 8.6
A fractional filling, %	37.1 ± 9.1	30.6 ± 11.5	37.8 ± 8.6
LAres, %	68.3 ± 30.2	25.0 ± 17.0 *	55.6 ± 37.3 *
LAmaxi, cm2/m2	9.1 ± 2.0	10.5 ± 2.3	9.8 ± 2.2
IMT, mm	0.6 ± 0.1	0.7 ± 0.1	0.6 ± 0.1

Ap: mitral peak A wave; a.u.: arbitrary units; BMI: body mass index; CV: cardiovascular; DAP: diastolic arterial pressure; Ep: mitral peak E wave; IVSd: end-diastolic interventricular septum thickness; HDL-C: high-density lipoprotein cholesterol; HOMA-IR: homeostatic model; HR: heart rate; IMT: carotid intima-media thickness; LAmaxi: left atrial indexed monoplane 4-chamber maximal area; LAres: left atrial reservoir function; LDL-C: low-density lipoprotein cholesterol; LV: left ventricular; LVEDHI: end-diastolic hypertrophy index; LVEDVI: end-diastolic volume index; LVEF: ejection fraction; LVESS: end-systolic stress; LVMI: mass index; LVPWd: end-diastolic posterolateral wall thickness; SAP: systolic arterial pressure; TC: total cholesterol; TG: triglycerides. * *p* < 0.05.

## Data Availability

Data are available upon request to the corresponding author. All the data are stored in a dedicated archive at Jilin Heart Hospital, Imaging Department, China.

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
