# Peer review of "Determinants of Left Atrial Compliance in the Metabolic Syndrome: Insights from the “Linosa Study”"

_jpm, 2022, doi:10.3390/jpm12071044_

Round 1
Reviewer 1 Report
The authors carefully revised the whole manuscript and replied to the questions/comments by reviewers. However, they were not able to provide parameters of echocardiography, partially due to the usage of small devices.
If they cannot provide these data, they should describe the reasons in the manuscript.
It is mutual consensus in our country that e', E/e', Tricuspid regurgitant velocity, and LA volume index are important to evaluate LV diastolic function. Please review the ESC guidelines and AHA guidelines for echocardiography to evaluate LV diastolic function and LA compliance.
Author Response
Reply
We would like to thank the Reviewer #1 for having carefully read our revised version of the manuscript. We agree with the reviewer’s comment. As suggested, we further explain and strengthen the limitations (see lines 452-459 high lightened in yellow).
Reviewer 2 Report
Dear Authors,
Thank you for your answers.
Thank you
Author Response
Reply
We would like to thank the Reviewer #2 for having carefully read our revised manuscript and for his/her positive assessment.
Reviewer 3 Report
Dear Editors,
during revision, the authors answered each issue of the reviewer. There are no further issues to comment on.
Author Response
Reply
We would like to thank the Reviewer #3 for having carefully read our revised manuscript and for his/her positive assessment.
Round 2
Reviewer 1 Report
The authors revised the manuscript according to my request. However, they did not have sufficient data of echocardiograpy to support their conclusion and described it as one of the limitations.
It is an interesting observation that "a direct independent influence of metabolic factors - in particular the lipid profile - on LA compliance" suggested by this study. Limitations of sample collection and lack of echography data seem to be a little too much.
Author Response
Reviewer 1 Round 2
We would like to thank the reviewer for his/her further comments.
Reviewer
The authors revised the manuscript according to my request. However, they did not have sufficient data of echocardiograpy to support their conclusion and described it as one of the limitations.
Reply
As a matter of fact, we did not state that the echocardiographic data provided were insufficient to support our conclusions. The conclusions of our study suggest an effect of metabolic factors on LA compliance, and the latter has been evaluated with a methodology that we have previously validated in a carefully designed animal study (reference #11: Barbier, P., Solomon, S. B., Schiller, N. B., and Glantz, S. A. Left atrial relaxation and left ventricular systolic function determine left atrial reservoir function. Circulation. 199910.1161/01.CIR.100.4.427.), which allowed to control for a number of confounding factors, and that established a relation between LA compliance (determined invasively) and LA reservoir function assessed by measuring LA area in the 4-chamber view. In this respect, the analysis of LA strain (which has a number of limitations – stressed by the current guidelines - the major being that the LA chamber is too thin and discontinuous at echocardiography to be adequately analysed by the current strain methodologies) should not be a prerequisite for the analysis of LA compliance.
In the Limitations paragraph, we mentioned that all the echocardiographic data requested by the current Guidelines to investigate LV diastolic function were not available. We did explain however that, since our patient sample was composed of subjects with normal LV anatomy and function, in this specific setting the analysis of transmitral flow was sufficient to discuss about the modalities of LV filling, even without knowledge of the tissue Doppler early diastolic e’ velocity. The latter parameter is used as an estimate of LV relaxation in heart disease, when LV anatomy, valvular function, and the diastolic transmitral flow profile are all normal, the LV relaxation is inevitably normal.
Reviewer
It is an interesting observation that "a direct independent influence of metabolic factors - in particular the lipid profile - on LA compliance" suggested by this study. Limitations of sample collection and lack of echography data seem to be a little too much.
Reply
As discussed in the Limitations paragraph, we believe that our population sample - although limited in number – carries the advantage of representing a unique closed population free from confounders. Certainly, further studies are needed to replicate our findings in larger and more diverse populations.
We discussed in the previous paragraph about the echocardiographic data.
This manuscript is a resubmission of an earlier submission. The following is a list of the peer review reports and author responses from that submission.
Round 1
Reviewer 1 Report
I had the pleasure of reviewing “Determinants of left atrial compliance in the metabolic syndrome: insights from the “Linosa Study” by Paolo Barbier et al. The authors associated left atrial parameters from echocardiography with the presence of metabolic syndrome. Both systolic blood pressure and total cholesterol/HDL ratio were associated with reduced LA compliance. LA functional parameters were also associated with increased all-cause mortality. Their theory that metabolic syndrome first leads to changes of the left atrial function before affecting the LV is very interesting.
While the statistic seems solid and most parts of the manuscript are well-written, I think that the authors drew to many conclusions from the data. They discuss that there would be an “effect of altered lipid profile on LV diastolic function” (line 337). However, this study is only observational and therefore only associations can be assumed. Furthermore, it is pity that no data about the development of cardiovascular disease in the population is presented. This would strengthen the link between LA function and cardiovascular disease. Considering the present data, only associations between metabolic syndrome and LA function can be made.
Minor comments:
- I would not consider the whole population totally “healthy” with a high prevalence of metabolic syndrome, hyperlipidaemia and hypertension. A better expression would rather be “apparently healthy”.
- Was data always normally distributed? Otherwise, non-parametric tests need to be used.
- Associations between LA function (+ other parameters) and death at 5 years may be included in the abstract.
Author Response
Reply to Reviewer #1
I had the pleasure of reviewing “Determinants of left atrial compliance in the metabolic syndrome: insights from the “Linosa Study” by Paolo Barbier et al. The authors associated left atrial parameters from echocardiography with the presence of metabolic syndrome. Both systolic blood pressure and total cholesterol/HDL ratio were associated with reduced LA compliance. LA functional parameters were also associated with increased all-cause mortality. Their theory that metabolic syndrome first leads to changes of the left atrial function before affecting the LV is very interesting.
Reply
We would like to thank the Reviewer #1 for having carefully read our manuscript and for his/her positive assessment.
While the statistic seems solid and most parts of the manuscript are well-written, I think that the authors drew to many conclusions from the data. They discuss that there would be an “effect of altered lipid profile on LV diastolic function” (line 337). However, this study is only observational and therefore only associations can be assumed.
Reply
We agree and rephrase accordingly. We also try to make both the “results” and “discussion” sections clearer and more balanced, as suggested (see corrections in red pages 4-14).
Furthermore, it is pity that no data about the development of cardiovascular disease in the population is presented. This would strengthen the link between LA function and cardiovascular disease. Considering the present data, only associations between metabolic syndrome and LA function can be made.
Reply
Correct, we add a sentence in the “limitations” (lines 452-454)
Minor comments:
- I would not consider the whole population totally “healthy” with a high prevalence of metabolic syndrome, hyperlipidaemia and hypertension. A better expression would rather be “apparently healthy”.
Reply
Thank you for the suggestion, we change accordingly in the sections “Abstract” (line 29), “Introduction” (line 61), “Materials and methods – study population” (line 69) and “Conclusions” (line 457).
- Was data always normally distributed? Otherwise, non-parametric tests need to be used.
Reply
Thank you, correct. We checked the statistical analysis according to the reviewer’s observation. We modify the paragraph accordingly (see lines 128-136).
Associations between LA function (+ other parameters) and death at 5 years may be included in the abstract
Reply
Thank you, it was missing. We now add a sentence in the abstract about the mortality at follow-up (lines 31-32).
Reviewer 2 Report
Barbier et al. adressed the correlation of determinants of the metabolic syndrome (metS) with those of left atrial (LA) compliance in 148 study participants of whom 41 had the metS. They found LA dysfunction in more than a half of the metS-population. This was mainly due to the subgroup of patients having arterial hypertension as one of the determinants of metS. The authors included a 5 year follow up. During this time, 24 participants had died. Of note, all deceased had a decreased LA reservoir function at time of enrolment. The strength of the study is in the inclusion of an “enclosed” population, namely the inhabitants of a small island in the Sicily Channel. Based on enrolment data, the subjects were treatment naïve or else showed highly dysregulated dyslipidemia and arterial hypertension. This nicely demonstrates the correlation of biometric data with the echocardiographic data of cardiac function and allows speculating about pathophysiologic mechanisms. So far missing in the manuscript is any treatment data. This should be added, as i.e. treatment with ACE-inhibitors or statins could influence the results. In addition, presenting data in figures or graphs would help to read the text which otherwise is loaded with many numbers and data.
Author Response
Reply to Reviewer #2
Reviewer
Barbier et al. addressed the correlation of determinants of the metabolic syndrome (MetS) with those of left atrial (LA) compliance in 148 study participants of whom 41 had the MetS. They found LA dysfunction in more than a half of the MetS-population. This was mainly due to the subgroup of patients having arterial hypertension as one of the determinants of MetS. The authors included a 5 year follow up. During this time, 24 participants had died. Of note, all deceased had a decreased LA reservoir function at time of enrolment. The strength of the study is in the inclusion of an “enclosed” population, namely the inhabitants of a small island in the Sicily Channel. Based on enrolment data, the subjects were treatment naïve or else showed highly dysregulated dyslipidaemia and arterial hypertension. This nicely demonstrates the correlation of biometric data with the echocardiographic data of cardiac function and allows speculating about pathophysiologic mechanisms.
Reply
We would like to thank the Reviewer #2 for having carefully read our manuscript and for his/her positive assessment.
So far missing in the manuscript is any treatment data. This should be added, as i.e. treatment with ACE-inhibitors or statins could influence the results.
Reply
Thank you. We apologize for missing such an important information. We add it (line 70). None of the enrolled subjects was on medications.
In addition, presenting data in figures or graphs would help to read the text which otherwise is loaded with many numbers and data.
Reply
Thank you for this suggestion, we add 2 figures and streamline the tables (tables 2 and 4, figures 1 and 4).
Reviewer 3 Report
Dear Authors,
Thank you.
Comments
1. Interesting analysis, but the number of patients in two groups is too smail to derive any safe conclusion.
2. Pts with MetS only 41 and without 107.
3. Very close population to adapt these results in general population. Many co-funding factors may affect the stud's results in general population
Thank you
Author Response
Reply to Reviewer #3
We thank the Reviewer #3 for having carefully read our manuscript.
- Interesting analysis, but the number of patients in two groups is too smail to derive any safe conclusion.
Reply
We agree. We tried to stress this limitation, now we add a further comment in the “conclusions” to further underline the need of larger studies to confirm our preliminary results (lines 462-463).
We still believe these results are worth presenting bearing in mind that what we consider a stimulating hypothesis, i.e. a link between dyslipidemia and left atrial chamber compliance, needs to be confirmed in a larger study.
- Pts with MetS only 41 and without 107.
Reply
Details about the Linosa Study are presented in reference #33 (Bellia et al.). In short, the overall adult population of Linosa Island consisted of 420 subjects at the time of enrolment. A total of 364 individuals agreed to participate in the original study. All participants were Caucasian. Complete data collection was available in 293 adults. For the aim of the present sub-study a healthy population could be considered. Thus, after exclusion of subjects with known disease, 148 healthy subjects were considered for the present sub-study, where prevalence of MetS is 27.7% (41 subjects), in keeping with reference # 33, where this higher prevalence than compared to the general Italian population (15-22%). We add some information in the paragraph “Study population” (lines 66-68). The weakness due to the small sample is underlined in the “Limitations”.
Very close population to adapt these results in general population. Many co-funding factors may affect the stud's results in general population
Reply
We agree with the reviewer's remark, as we stated in the "Limitations". The results of the Linosa Study cannot be directly translated to the Italian population or to any general population, mainly due to possible genetic differences, to which the higher prevalence of MetS could be related. However, we think that this line of reasoning should not distract from our aims and results, more specifically the hypothesis of a direct effect of the lipid profile on the thin left atrial chamber.
Reviewer 4 Report
The idea of this research is interesting, but the study population is too small. If this population represents a whole country, the authors should provide the evidence.
It is very confusing if the authors are focusing on LA diastolic function or LV diastolic function. Figure 1 and Figure 2 are the analysis of LV data, and most of the parameters of Table 2, 4, and 5 are LV data. If the authors want to analyze LV diastolic function, they should provide e’, E/e’, Tricuspid regurgitant velocity, and LA volume index.
LA volume index should be presented to analyze LA compliance.
Mortality at 5 years was analyzed, but the sample size was too small to make conclusion as the authors admitted. Only 4 people died of CVD out of 148 participants. There is not much to say about mortality.
Discussion section is hard to read and understand. There are too many quotations listed in random order.
It sounds that the authors tried to say hyperlipidemia of MetS causes decreased LA compliance which may lead to CAD. In my own experience, patients diagnosed with MetS in health check-up usually have “overlooked” or “unrecognized” hypertension. It is well-known that hypertension causes ventricular diastolic dysfunction and LA dilatation. If the study population of this research had “in-diagnosed” hypertension at enrollment, it is not surprising that the patients showed ventricular diastolic dysfunction and/or LA dilatation.
Author Response
Reply to Reviewer #4
We thank the Reviewer #4 for having carefully read our manuscript.
The idea of this research is interesting, but the study population is too small. If this population represents a whole country, the authors should provide the evidence.
Reply
Thank you. We add some additional information in the study population section (lines 66-68) to better define the “Linosa study” population. We agree that the small sample represents a limitation, as we state, we try to define this limitation together with the uniqueness of the sample (lines …).
As also replied to Reviewer #3, we do believe that this study suggests an intriguing hypothesis, i.e. a direct effect of the lipid profile on the thin left atrial chamber. There is a clear need for larger studies to confirm it, as we stress in the limitations paragraph.
it is very confusing if the authors are focusing on LA diastolic function or LV diastolic function. Figure 1 and Figure 2 are the analysis of LV data, and most of the parameters of Table 2, 4, and 5 are LV data. If the authors want to analyse LV diastolic function, they should provide e’, E/e’, Tricuspid regurgitant velocity, and LA volume index.
Thank you for these helpful comments. As known, LA function and LV diastolic function are intertwined, therefore, they cannot be analysed separately. In our study, the analysis of LV diastolic function was essential to determine that the reduction in LA compliance was not related, as confirmed by multiple regression analysis, to LV diastolic dysfunction. This is important, as an impaired LV relaxation is a major determinant of LA remodeling. Our results suggest that in patients with MetS and dyslipidaemia, there might be a direct effect on LA function, with a direct mechanism - intramyocardial infiltration? –, as has been shown in amyloidosis, where severe LA dysfunction has been found related to the intramyocardial deposition of amyloid.
We also agree with the reviewer that an accurate analysis of LV diastolic dysfunction should be conducted according to current Guidelines, which require the use of Doppler tissue imaging and quantitative analysis of pulmonary systolic pressure and LA volume. We could not exploit the use of tissue Doppler to directly assess LV relaxation (with e’), because the Linosa study was conducted utilizing a portable equipment (back in 2008), which served the purpose of screening the population of Linosa. However, taking into account that our population was apparently healthy (asymptomatic without any history of CVD), that both LV dimensions and systolic function were normal, that the right heart showed no alteration at qualitative analysis, and that LA dimensions (albeit measured as monoplane area index and not biplane volume index) were normal, the probability of the presence of alterations in LV relaxation or in LV chamber compliance was rather low, also excluded by the analysis of the transmitral flow profile. In a subject with a normal LV diastolic function (both relaxation and chamber compliance parameters), most of the stroke volume enters the ventricular cavity in early diastole (the fractional filling parameter E in our study), while atrial contraction completes the LV filling to obtain an optimal end-diastolic volume. When relaxation is prolonged, early filling is impaired with a reduced early filling and prolonged E wave deceleration time, and late diastolic filling is increased. The transmitral velocity profile was normal in our patients (we did not include in the Tables the deceleration time, which was normal and mainly dependent on age, as expected in a healthy population. We could detect an initial alteration of the mitral velocity profile only in the subjects of Group 4 (MetS and hypertension). Finally, in an asymptomatic population with normal LV and LA dimensions, normal LV systolic function and stroke volume, and normal mitral flow velocity profile, LV compliance can also be considered normal. We try to summarize these concepts in the discussion (see pages 11-14)
LA volume index should be presented to analyse LA compliance.
We fully agree with the reviewer and the Guidelines; LA volume is the best parameter for analyzing LA size and derived parameters in the general population. Unfortunately, this measurement was not available, however, as just discussed above, all subjects showed normal LA anatomy, and therefore we can assume that the LA area in this specific situation was sufficient for the purpose of the analysis.
Mortality at 5 years was analyzed, but the sample size was too small to make conclusion as the authors admitted. Only 4 people died of CVD out of 148 participants. There is not much to say about mortality.
Reply
True, the mortality data may only suggest the association between atrial dysfunction and increased CV risk, we add a comment in the” Limitations” (lines 453-455).
Discussion section is hard to read and understand. There are too many quotations listed in random order.
Thank you. We apologise for the confusion. According to the Reviewer’s comments, we re-write the Discussion, and revise the Results section, adding 2 figures (Figure 2 and 4) and streamlining table 2 and 4, to make our findings more readable and understandable. Furthermore, the entire manuscript has been revised by a native speaker, as stated in the Acknowledgments (lines 465-467).
It sounds that the authors tried to say hyperlipidemia of MetS causes decreased LA compliance which may lead to CAD. In my own experience, patients diagnosed with MetS in health check-up usually have “overlooked” or “unrecognized” hypertension. It is well-known that hypertension causes ventricular diastolic dysfunction and LA dilatation. If the study population of this research had “in-diagnosed” hypertension at enrolment, it is not surprising that the patients showed ventricular diastolic dysfunction and/or LA dilatation.
Thank you. Again, we apologize for the confusion. We cite results from the literature: “LA remodeling has gained acceptance as a marker which can detect the presence of cardiovascular disease”. We agree that there is no evidence of a link between LA compliance and coronary artery disease. In fact, in the Conclusions we suggest that our results reinforce “the important prognostic role of the left atrium in cardiovascular disease and strengthen the need for accurate and strict lipid control strategies”. Regarding the detection of MetS and Hypertension in an apparently healthy population, we agree with the reviewer’s comments. It is interesting to observe that in the Linosa population we analysed, the presence of underdiagnosed MetS and hypertension had affected LA compliance independently from LV diastolic dysfunction.